# What Does Sustainability Mean? Perceptions of Future Professionals across Disciplines

**Andrea Beatriz Damico [1,2,*]** , **José María Aulicino [2]** and **Jorgelina Di Pasquale [1]**

1    Faculty of Veterinary Medicine, University of Teramo, Piano D'Accio, 64100 Teramo, Italy
2    Faculty of Agricultural Science, National University of Lomas de Zamora, Ruta Provincial 4 Km 2, Llavallol, Buenos Aires B1836, Argentina
*    Correspondence: abdamico@unite.it

**Abstract:** Given the negative externalities of the traditional productive system, sustainable development has become a productive alternative that attempts to improve the quality of life of present and future generations. The aim of this research was to understand the degree of perception and knowledge of sustainability of university students attending different courses, who represent future Argentinian professionals. A survey was conducted on a representative sample of those enrolled in the faculties of Agricultural, Economic, and Social Sciences of the National University of Lomas de Zamora, Buenos Aires. The results showed that, in terms of awareness, the environmental dimension of sustainability stands out above the others. Sustainability involves preserving natural resources, favoring biodiversity, reducing environmental risks, and finding a balance between the development of humanity and care for the environment. Only 10% of the respondents identified the three sustainability dimensions contemporaneously, and most of these individuals were not informed through university courses. No substantial differences were found in the knowledge of sustainability among students of different faculties. Universities, as trainers of professionals and leaders, should further develop the subject in their curricula, to improve knowledge of sustainability, so that graduates can better face future professional challenges. Similarly, students should strive to know about sustainability and its components to defend and improve it in all areas of work.

**Keywords:** sustainable development; environmental sustainability; economic sustainability; social sustainability; students' perception; students' knowledge; change management; sustainability future professionals; education; Argentinian professionals

## 1. Introduction

Sustainable development has emerged as an alternative means to address the damage caused to the environment by the overexploitation of resources, environmental degradation, climate change, and population increase [1]. According to the UN [2], by 2050, world population growth will increase by 26%, to a total of 9.7 billion people. This growth will affect the quality of life of future generations. Thus, the need arises for a "development that meets the needs of the present without compromising the ability of future generations to meet their own needs" [1], which can be applied as a concept in almost all disciplines.

The term "sustainability" refers to the way of defending, applying, or arguing in favor of sustainable development. According to the United States Environmental Protection Agency [3], sustainability is a concept that can be defined in different ways, but its principles remain constant: balancing a growing economy, protection of the environment, and social responsibility [4].

Today, sustainability has become a popular term [5], but it is also a broad, complex, and multidisciplinary topic [6,7], which includes different dimensions, such as the environmental, social, and economic dimensions mentioned in United Nations Resolution A/60/1 of 2005 [8]. Similarly, it is the most widely referenced term in the literature ([5,7,9,10] among others).

Several authors have recognized that, when sustainability is mentioned, people generally refer exclusively to the environmental dimension [11–14], especially due to the influence of the media on issues related to the environment [11], while the social and economic dimensions are relegated in many cases [11,12,15,16].

Therefore, to promote sustainable development and sustainability, decision-makers, policymakers, and the scientific and educational communities should act with a broader vision, encompassing all dimensions of sustainability, and develop creative, ethical, trained, and competent thinkers [11] who will improve living conditions for all citizens.

Within this overall concept, education is a strategic factor in the move toward sustainable production around the world [17]. The creation of The Higher Education Sustainability Initiative (HESI), launched before the Rio + 20 Conference in 2012 [18], demonstrates the importance of training at the international level. This initiative brings together various entities of the United Nations and higher education communities, sharing knowledge and offering training to advance in the field of sustainability.

Universities can play a key role in the training of future professionals [19], thanks to the fact that they have a significant impact on society [20]. Some authors have tried to understand how university students from different disciplines perceive sustainability [11,12,15,19,21–23] and how much they know about it [4,5,24,25].

Through a study of the discourse of university students (*n* = 1889) in the United Kingdom, Kagawa [12] highlighted that only a third of students were very familiar with the term "sustainable development", whereas more than 70% of respondents indicated that sustainability was a "good thing". In addition, the author reported that environmental sustainability was the most widely recognized dimension.

Emanuel and Adams [23] carried out a comparative study of students (*n* = 406) from two universities in the USA which showed that more than 30% of students had little knowledge about sustainability and that less than 20% had substantial knowledge about the subject. In their research of university students (*n* = 1000) conducted in Germany, Barth and Timm [25] highlighted "sophisticated" knowledge about sustainability, and, like Kagawa [12], reported that the environmental aspect of sustainability stood out above the other dimensions. In addition, they considered this approach "important" or "very important" (28.5% and 35.7%, respectively) for their professional and private lives.

Similarly, in a study of students and university leaders (*n* = 1134) conducted in China, Yuan and Zuo [22] identified that the environmental dimension was perceived as most important for sustainable development, with general agreement that environmental aspects of sustainability should be given higher priority over other dimensions. Similarly, in a paper on US students (*n* = 82), Watson et al. [15] found that the students focused mainly on the environmental dimension, as compared with the economic and social dimensions.

In a study of US university students (*n* = 1389), Zwickle et al. [4] detected a medium–high assessment (69%) of the importance of sustainability. By considering the three dimensions separately, students scored similarly on environmental (73%) and economic (71%) questions and obtained lower scores (61%) on social questions. In the research of Msengi et al. [5], also involving US university students (*n* = 73), it was found that only a minority of respondents knew what sustainability was, but 95.8% indicated that it was something important.

Although sustainability is an abstract concept [4,26], students generally think it is something "very important", "good" or "positive" [5,11,12]; even when they are distrustful of the concept itself, they seem to have a positive attitude towards the core components of sustainability [26]. However, knowledge of sustainability is partial, perhaps superficial, and not always consistent with the level of knowledge expected to promote the various production activities which will achieve the much-desired safeguarding of the planet. Among such useful activities to be promoted, universities should strive to educate students for a more sustainable future [5,27] and evaluate the knowledge so attained [24]. To this end, it will be necessary to incorporate sustainability into the curricula at all academic levels, with a focus on the balance between the three dimensions [28]. The overall theme

must be inclusive, to improve knowledge about sustainable production, in order to support all productive sectors and improve all supply chains.

Students are ideally positioned to develop skills and capabilities in creative and critical problem solving for their future professional practice [11,15,29]. They need to understand how their decisions and actions can affect the environment, the economy, and society as a whole [15,30].

To the best of our knowledge, in the international literature, few studies have explored the perception and knowledge of students attending courses that correspond to the dimensions of sustainability, and no studies have explored the situation in Argentina. Understanding how Argentinian university students perceive sustainability and its dimensions is important, since they will soon be integrated into the workplace and have the opportunity to make decisions that provide solutions consistent with the concept of sustainability [31]. Therefore, the objective of this work was to determine the perception and knowledge of sustainability, and its dimensions, among students from different faculties who will become professionals in the future. To incorporate the three dimensions of sustainability, this research focused on university students at the National University of Lomas de Zamora, Buenos Aires, Argentina, from three of its faculties: Agricultural Sciences (environmental dimension), Economic Sciences (economic dimension), and Social Sciences (social dimension).

The hypothesis of this research was that the students of the different faculties would have higher levels of perception and knowledge about the dimension of sustainability most closely associated with their chosen university course. This study contributes to filling a gap in the literature by determining how much the students analyzed knew about sustainability, and by highlighting the possibility of intervening in their training, in order to provide them with the necessary tools to adapt their professional development to the challenges of a sustainable future society.

## 2. Materials and Methods

Between the months of April and July 2021, at the National University of Lomas de Zamora, Buenos Aires, Argentina, a survey was conducted, using Computer Assisted Web Interviewing (CAWI) methodology, among students of the faculties most widely related to the three areas of sustainability, i.e., the faculties of Social Sciences, Agricultural Sciences, and Economic Sciences. The CAWI methodology was chosen in order to reach all the students enrolled in the three degree courses involved in the survey and thus obtain a representative sample of the target population. The purpose was to analyze the knowledge and perception of "sustainability" and its dimensions. The respondents were over 18 years of age, but no age limit was established.

The survey was conducted through a Google Form, and the types of questions were open, closed, and semi-closed, with answers on metric scales (from 0 to 10) and categorical.

A pre-test and a subsequent pilot test of the form were carried out among a small group of students ($n = 20$) belonging to the target population, to verify that the questionnaire met the study objectives, and to detect inconsistencies, repetitions, and sequencing errors in the questions. Similarly, the wording, the fluidity of the survey, and the understanding of the respondents were also tested [32].

For the online dissemination of the questionnaire, the authorities of each faculty collaborated by sending it to the students through the institutional mailing list. Participation in the survey was voluntary and the participants signed an informed consent form.

Random probabilistic sampling was performed on the nearly 20,000 students of the faculties of Social Sciences, Agricultural Sciences, and Economic Sciences (approximately 10,000, 1500, and 8000 students, respectively). We worked with a representative sample, with a confidence level of 95%, a margin of error of 5%, and heterogeneity of 50%.

The survey consisted of 35 questions organized into four thematic sections:

Section 1—Demographic and personal characterization of the sample: personal, demographic, and university information was requested.

Section 2—Perception of sustainability and its dimensions: an open question was asked in which the respondents freely reported the first word or phrase which they associated with the concept of sustainability.

Section 3—Importance attributed to sustainability and its dimensions: different statements about sustainability were proposed and respondents were asked to assess them on a scale between 0 and 10 according to the degree of importance attributed. Subsequently, they were asked about the three dimensions (environmental, social, and economic), to find out to what extent they were aware of such dimensions.

Section 4—Knowledge of sustainability and its dimensions: without asking directly about the definition of sustainability due to its breadth and evolution, the participants were asked to assess their degree of agreement with some proposed phrases related to sustainability. Likewise, respondents were asked to self-assess their own knowledge of the subject analyzed. In addition, they were requested to report the sources of information they used and assess the relevance of the topic to their future professional development.

Microsoft Excel was used for the management of the database. For the qualitative computer-aided analysis (CAQDAS), the NVivo 12 Edition Plus software of QSR International [33] was used; and for quantitative analysis, Infostat Version 2020 software was used [34].

### 2.1. Discourse Analysis

Section 2 of the questionnaire consisted of an open question, in which respondents expressed themselves freely about sustainability. This section was analyzed by using a mixed lexicometric approach (quali-quantitative), based on the techniques developed by the French School of Data Analysis (Analyse des Données) by Benzécry and collaborators [35,36]. To achieve this, a verbatim database of the students' answers was created, and these expressions were then consolidated in terms of their semantics and spelling. Subsequently, the "frequency of words" ($f$p) was analyzed to achieve a global view and identify the words that were most frequently repeated within the discourse. The identified words were required to have a length greater than three letters. In addition, "derived words" (words with the same root grouped by the software) and "synonyms" were considered. In the procedure, those words considered "empty of meaning" and those which did not represent concepts referring to the type of analysis to be carried out (articles, prepositions, common verbs, etc.) were eliminated. The general discourse of each faculty were analyzed, to see if there were differences between them.

### 2.2. Descriptive Analysis and Bivariate Analysis

Sections 3 and 4 of the questionnaire were analyzed through descriptive analysis and frequency tables, in order to characterize the sample demographically, and to obtain general information. Subsequently, quantitative statistical analyses were carried out to verify the existence of differences in perceptions and knowledge of sustainability associated with the students' faculties. To this end, bivariates were used, as they make it possible to establish relationships between pairs of variables to determine the statistical significance of the differences observed [37].

## 3. Results

### 3.1. Demographic and Personal Characterization of the Sample

The sample was composed of 1063 students and their demographic characterization is presented in Table 1. In terms of age, 46.7% were 28 years of age or older. It is common in Argentina for the student population to both study and work, and high percentages of working students (between 67.5% and 75.1%) were identified in the three faculties analyzed (Table 1).

**Table 1.** Demographic characteristics of the respondents.

| Faculty | Agricultural Sciences *n* = 321 | Social Sciences *n* = 374 | Economic Sciences *n* = 368 | Total No. of Students *n* = 1063 |
|---|---|---|---|---|
| Gender (%) | | | | |
| Female | 60.1 | 87.1 | 61.9 | 70.3 |
| Male | 39.9 | 12.9 | 38.1 | 29.7 |
| Age (%) | | | | |
| 18–27 years old | 55.1 | 46.0 | 58.8 | 53.2 |
| 28–40 years old | 31.5 | 36.4 | 30.9 | 32.7 |
| 41 + years old | 14.3 | 17.6 | 10.3 | 14.0 |
| Students who work (%) | | | | |
| Yes | 75.1 | 71.9 | 67.5 | 71.3 |
| No | 24.9 | 28.1 | 32.5 | 28.7 |

*3.2. Perception of Sustainability and Its Dimensions*

The free discourse on sustainability yielded a total of 2067 words. After these were purified and verified semantically and orthographically, the total was reduced to 1852 words. Then, by removing the words of three or fewer letters and those empty of meaning, a consolidated total of 1449 words was obtained. The frequency of these words ($fp$) was then analyzed; that is, we counted the number of times each word was repeated. By these means, we identified a total of 261 different words.

Within each faculty, the Agricultural Sciences students produced a greater diversity of discourse with 510 words and an $fp$ of 147, followed by the Social Sciences students with 494 words and an $fp$ of 152, and. lastly, the Economic Sciences students, with 445 words and an $fp$ of 126.

Below, in rank order, are the 31 words with 10 or more counts (repetitions), equivalent to 65.4% of the consolidated speech (Table 2).

**Table 2.** Consolidated discourse words with 10 or more counts.

| N° | Word | Count | Similar Words in Terms of Root or Synonyms |
|---|---|---|---|
| 1 | Environment | 118 | environment, environmental |
| 2 | Ecology | 88 | ecology, ecologic, agroecology |
| 3 | Resources | 64 | resource, resources |
| 4 | Care | 61 | care, caring, take care |
| 5 | Sustainable | 54 | support, sustain, sustainable, sustainability, sustained, self-sustainable, self-sustain, sustained |
| 6 | Future | 53 | future |
| 7 | Recycling | 52 | recyclable, recycled, recycle |
| 8 | Time | 40 | time |
| 9 | Natural | 34 | natural, nature |
| 10 | Economy | 34 | economy, economic |
| 11 | Development | 32 | develop, development |
| 12 | Balance | 32 | balance, balanced |
| 13 | Responsibility | 26 | responsibility, responsible |
| 14 | Awareness | 24 | awareness, aware |
| 15 | Long | 20 | long |

**Table 2.** *Cont.*

| N° | Word | Count | Similar Words in Terms of Root or Synonyms |
|---|---|---|---|
| 16 | Maintain | 18 | maintain, maintenance |
| 17 | Long-term | 18 | long-term |
| 18 | Friendly | 17 | friendly |
| 19 | Lasting | 17 | last, lasting |
| 20 | Capacity | 15 | capacity |
| 21 | Life | 15 | life |
| 22 | Use | 14 | use |
| 23 | Need | 13 | need, needs |
| 24 | Work | 13 | works, work, job |
| 25 | Planet | 12 | planet |
| 26 | Good | 11 | good |
| 27 | Better | 11 | better, improvement, improve |
| 28 | Renewable | 11 | renewable, renovation |
| 29 | Pollution | 11 | pollution, pollutant |
| 30 | Benefit | 10 | benefit, beneficial |
| 31 | Well-being | 10 | well-being |

These 31 words were grouped according to thematic affinity, resulting in four emerging categories related to the following themes:

1. Environmental: environment, ecology, care, natural, friendly, planet, good, and pollution, resulting in 352 total counts.
2. Social: responsibility, awareness, life, needs, and well-being, resulting in 101 total counts.
3. Economic: economy and capacity, resulting in 49 total counts.
4. Crosscutting words (words assignable to any of the topics above): resources, sustainable, recycling, development, balance, maintain, use, good, better, renewable, benefit, and all words related to time (future, time, long, term and durable), resulting in 446 total counts.

Excluding cross-cutting words, the environmental issue ranked first, with 70.1% of the words used. This was followed, by some distance, by the social issue with 20.1%, while the economic issue was relegated to third place with 9.8% of the words used (Figure 1).

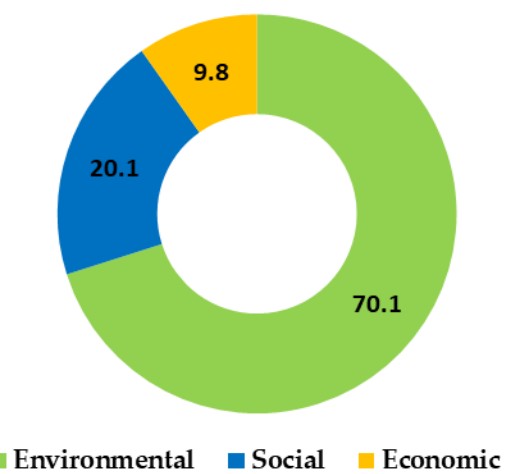

**Figure 1.** Word count by thematic areas (%).

In our analysis of the 31 words, the highest number of words (329) was used by students of Agricultural Sciences, closely followed by those of Social Sciences (326) and, finally, Economic Sciences (293), i.e., the same order as before.

Not counting the cross-cutting themes, all students related sustainability in the first place with environmental issues, while social and economic issues were not as frequently mentioned, with a wide distance between the first and the other two themes (Table 3).

**Table 3.** Total number of words by dimension and by faculty.

| Dimensions | Total No. of Words | No. of Words per Faculty | | |
|---|---|---|---|---|
| | | Agricultural Scs. | Economic Scs. | Social Scs. |
| Environmental | 352 | 105 | 127 | 127 |
| Social | 101 | 38 | 25 | 25 |
| Economic | 49 | 14 | 20 | 20 |

As shown in the following figures, the words mentioned 10 or more times were as follows: Agricultural Sciences students: 14 words (Figure 2); Economic Sciences students: 10 words (Figure 3); Social Sciences students: 12 words (Figure 4).

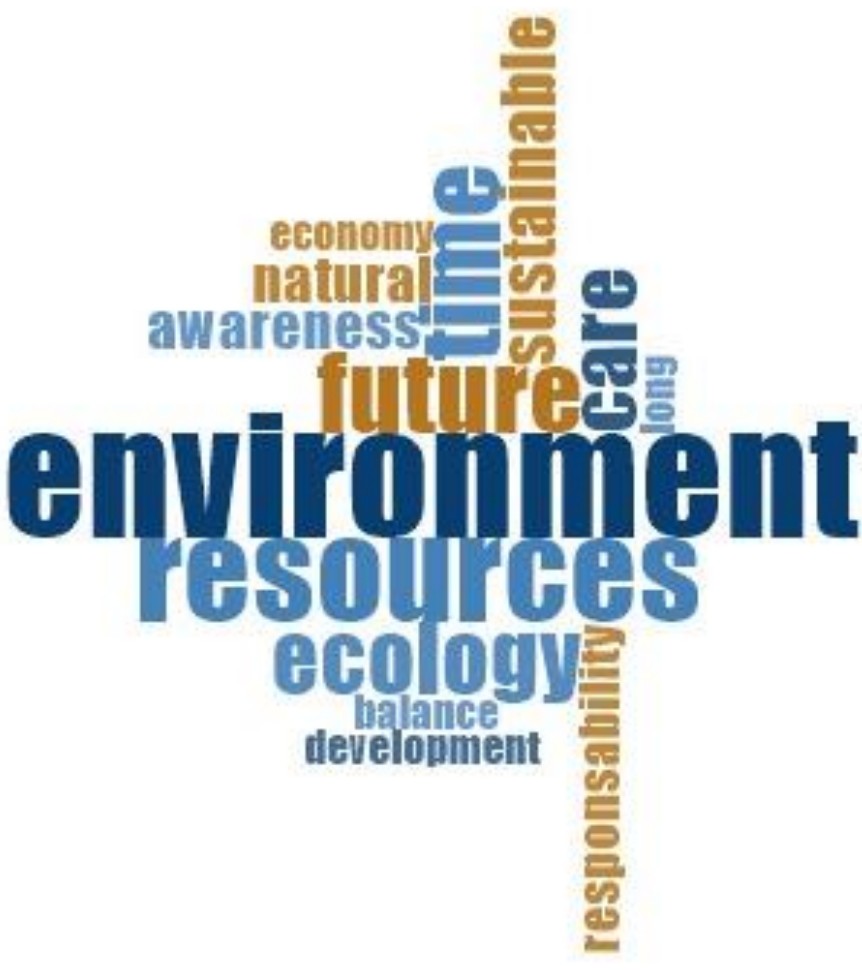

**Figure 2.** Frequency of words of Agricultural Sciences students.

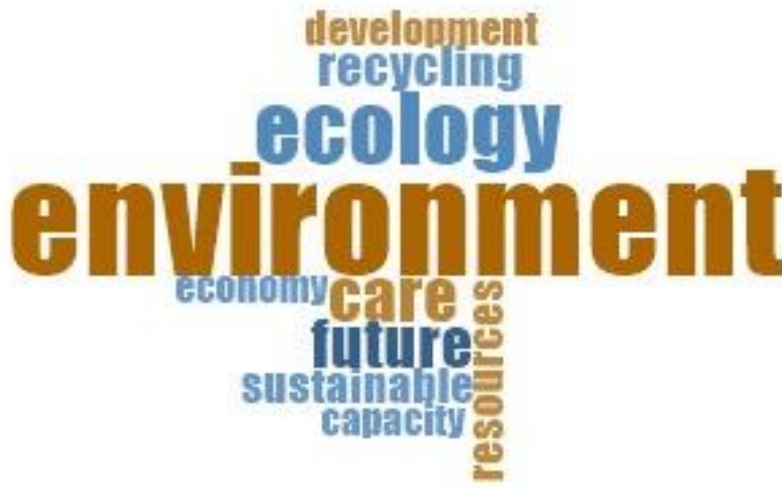

**Figure 3.** Frequency of words of Economic Sciences students.

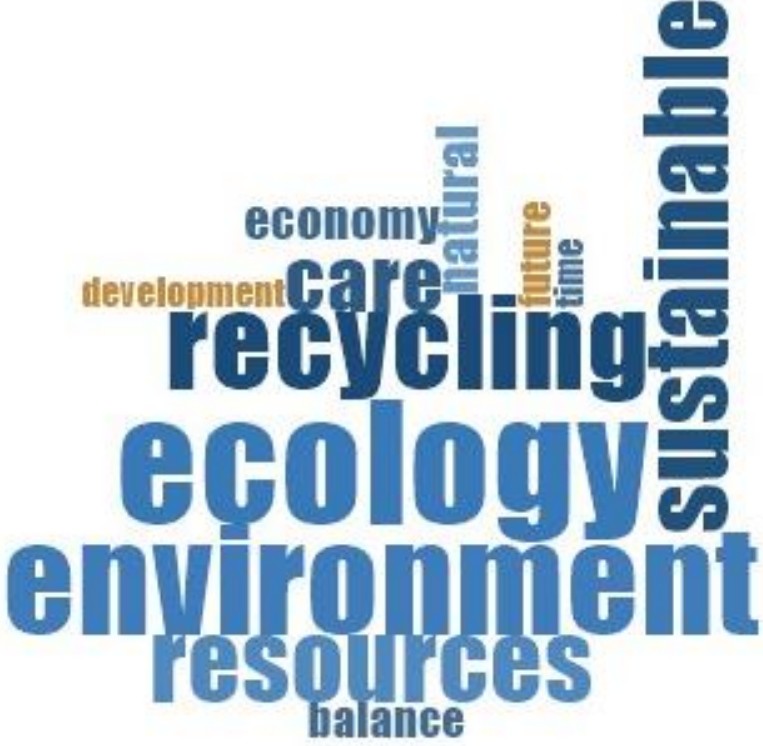

**Figure 4.** Frequency of words of Social Sciences students.

Comparing the words used by the respondents of the three faculties (Figure 5), we found that the expressions used seemed quite similar in lexical terms. Despite the difference in their academic backgrounds, the respondents used practically the same vocabulary. The most frequently mentioned word in the Faculty of Economic Sciences was "environment" ($f$p = 54), followed by "ecology" ($f$p = 32). In the discourse of Social Sciences students, the same words appeared in the first two places, but in reversed positions, with "ecology" ranking first ($f$p = 36), followed by "environment" ($f$p = 32). Among students of Agricultural Sciences, "environment" was the first-ranked word with 32 counts, but "resources" appeared in second place ($f$p = 27), while "ecology" only appeared in fifth place ($f$p = 20), perhaps due to its more specific usage in environmental issues.

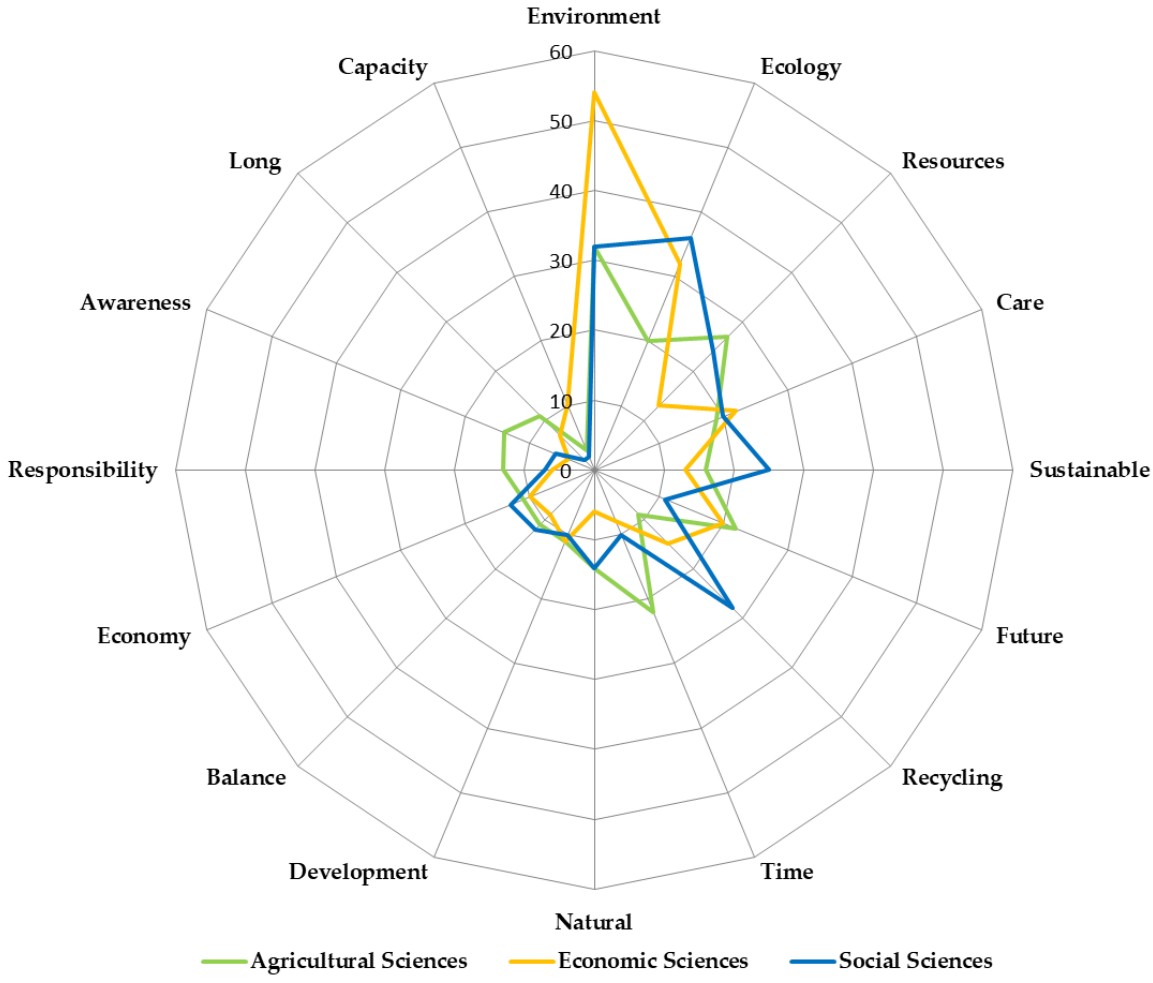

**Figure 5.** A comparison of words with 10 or more counts in any of the faculties.

### 3.3. Importance Attributed to Sustainability and Its Dimensions

Interviewees were asked to rate the level of importance attributed to each of the 12 proposed affirmative statements about sustainability. These are presented in Table 4, ordered by theme.

A total of 65% of the students valued the 12 proposed statements as very or extremely important. In particular, more than 90% of the students assigned values higher than 9 to statement 4 (related to social issues), and to statements 5, 6, and 8 (related to environmental issues).

For each of the statements, we analyzed whether there were significant differences between the evaluations of the students of the three faculties. In 10 of the 12 statements, there were no significant differences ($p > 0.05$), but there was a significant difference for statement 4 ("It requires more training and work to reduce the human impact on the environment") ($p = 0.005$) and for statement 12 ("It strives to reduce losses to make more efficient use of resources") ($p = 0.02$). Concerning statement 4, through a comparison of pairs, it transpired that the students of Economic Sciences assigned intermediate importance to sustainability while those of Agricultural Sciences considered it to be of little importance. On the other hand, in the case of statement 12, the students of Economic Sciences and those of Social Sciences both attributed moderate importance to sustainability, while Economic Sciences students valued it at the extremes, as either very important or not important at all.

**Table 4.** Assessment of the degree of importance attributed to each statement with respect to sustainability.

| Theme | Statement on the Importance of Sustainability | Median | Faculty | Median | Little or Not Important (*) | Moderately Important (**) | Very or Extremely Important (***) |
|---|---|---|---|---|---|---|---|
| | | | | | Students' Evaluation | | |
| | | | | | As Percentage (%) | | |
| **Social** | 1. Allows social development and roots in the local territory. | 8.14 | Agricultural | 8.23 | 3.1 | 15.0 | 81.9 |
| | | | Economic | 7.94 | 2.8 | 16.3 | 80.9 |
| | | | Social | 8.24 | 2.4 | 13.9 | 83.7 |
| | 2. Respects the human rights of producers and workers. | 7.90 | Agricultural | 7.76 | 6.5 | 20.6 | 72.9 |
| | | | Economic | 7.79 | 6.6 | 19.4 | 74.0 |
| | | | Social | 8.14 | 6.1 | 13.4 | 80.5 |
| | 3. Protects the public health of citizens. | 8.6 | Agricultural | 8.56 | 2.2 | 12.1 | 85.7 |
| | | | Economic | 8.52 | 3.4 | 11.6 | 85.0 |
| | | | Social | 8.62 | 2.7 | 8.6 | 88.8 |
| | 4. Requires more training and work to reduce the human impact on the environment. | 9.03 | Agricultural | 9.07 | 3.1 | 3.1 | 93.8 |
| | | | Economic | 9.07 | 0.0 | 6.2 | 93.8 |
| | | | Social | 8.95 | 1.6 | 6.1 | 92.2 |
| **Environmental** | 5. Balances the development of humanity and care for the environment. | 9.19 | Agricultural | 9.32 | 0.31 | 4.36 | 95.3 |
| | | | Economic | 9.17 | 0.9 | 4.7 | 94.4 |
| | | | Social | 9.09 | 1.6 | 4.3 | 94.1 |
| | 6. Maintains natural resources over time, for present and future generations. | 9.31 | Agricultural | 9.35 | 1.2 | 4.4 | 94.4 |
| | | | Economic | 9.30 | 0.3 | 4.1 | 95.6 |
| | | | Social | 9.29 | 0.3 | 3.5 | 96.3 |
| | 7. Adopts low-polluting production processes (less use of chemicals). | 8.83 | Agricultural | 8.93 | 1.6 | 4.7 | 93.8 |
| | | | Economic | 8.76 | 1.9 | 9.1 | 89.0 |
| | | | Social | 8.82 | 2.4 | 7.8 | 89.8 |
| | 8. Favors biodiversity, reduces environmental risks (e.g., erosion, floods, fires). | 9.05 | Agricultural | 9.23 | 0.6 | 3.7 | 95.6 |
| | | | Economic | 8.99 | 1.6 | 5.9 | 92.5 |
| | | | Social | 8.94 | 1.9 | 7.8 | 90.4 |
| **Economic** | 9. Is easier to implement on small production scales (e.g., family farming). | 7.94 | Agricultural | 8.02 | 4.3 | 18.1 | 77.6 |
| | | | Economic | 7.66 | 5.9 | 20.9 | 73.20 |
| | | | Social | 8.14 | 2.4 | 19.0 | 78.6 |
| | 10. Needs more labor than the traditional one. | 7.22 | Agricultural | 7.21 | 10.3 | 22.4 | 67.3 |
| | | | Economic | 6.90 | 10.9 | 27.8 | 61.3 |
| | | | Social | 7.54 | 7.2 | 23.8 | 69.0 |
| | 11. Is a profitable activity that generates jobs. | 8.48 | Agricultural | 8.64 | 0.9 | 8.7 | 90.3 |
| | | | Economic | 8.28 | 2.8 | 13.8 | 83.4 |
| | | | Social | 8.51 | 1.3 | 11.8 | 86.9 |
| | 12. Strives to reduce losses to make more efficient use of resources. | 8.31 | Agricultural | 8.37 | 2.8 | 11.5 | 85.7 |
| | | | Economic | 8.30 | 4.7 | 10.0 | 85.3 |
| | | | Social | 8.25 | 1.9 | 15.8 | 82.4 |

Note: The students' evaluations correspond to: (*) Little or not important (scores between 0 and 3), (**) Moderately important (scores between 5 and 7), (***) Very or extremely important (scores between 8 and 10).

Regarding the set of four statements belonging to each of the three dimensions of sustainability, it is evident that the students assigned different degrees of importance (Table 5). They valued the environmental dimension as most important (Median = 9.10), followed by the social dimension (Median = 8.41) and, finally, the economic (Median = 7.98).

There were no significant differences in the assessment made by the students of the three faculties as regards the social dimension ($p = 0.55$) and the environmental dimension ($p = 0.21$). On the other hand, there was a significant difference in assessment of the economic dimension ($p = 0.03$). The students of Agricultural Sciences attributed a higher value to the economic dimension than did the students of Economic Sciences, with the latter valuing this dimension to an intermediate and low extent.

**Table 5.** Assessment of the degree of importance of each Sustainability Dimension.

| Dimension | Median | Faculty | Median | Students' Evaluation | | |
|---|---|---|---|---|---|---|
| | | | | Little or Not Important (*) | Moderately Important (**) | Very or Extremely Important (***) |
| | | | | As Percentage (%) | | |
| Social | 8.41 | Agricultural | 8.41 | 3.7 | 12.7 | 83.6 |
| | | Economic | 8.33 | 3.2 | 13.4 | 83.4 |
| | | Social | 8.49 | 3.2 | 10.5 | 86.3 |
| Environmental | 9.10 | Agricultural | 9.2 | 0.9 | 4.3 | 94.8 |
| | | Economic | 9.05 | 1.2 | 5.9 | 92.9 |
| | | Social | 9.03 | 1.5 | 5.8 | 92.7 |
| Economic | 7.98 | Agricultural | 8.06 | 4.6 | 15.2 | 80.2 |
| | | Economic | 7.79 | 6.1 | 18.1 | 75.8 |
| | | Social | 8.11 | 3.2 | 17.6 | 79.2 |

Note: The students' evaluations correspond to (*) Little or not important (scores between 0 and 3), (**) Moderately important (scores between 5 and 7), (***) Very or extremely important (scores between 8 and 10).

### 3.4. Knowledge of Sustainability and Its Dimensions

The surveyed students were able to identify the dimensions that make up sustainability from a set of different proposed dimensions. In particular, the environmental dimension was recognized by 96.2% of those surveyed, followed by the economic dimension which was identified by 83.5% and the social dimension by 80.3%. Unfortunately, all three dimensions were simultaneously identified by less than 10% of the respondents (9.4% = 100 students). There were no significant differences in the relative frequencies ($p = 0.69$) with which the students of the different faculties were able to correctly identify the three dimensions together.

Without addressing the definition of sustainability expressly, due to its breadth and evolution, an attempt was made to understand how much the students knew about sustainability. They were asked to provide their degree of agreement with some proposed statements (Figure 6). Ninety percent of the students strongly agreed that "sustainability is essential to preserve resources since current productions are unsustainable in the long term". 54% of the students also strongly agreed that "it is a concern of consumers that will change the way of producing, and it is not a marketing strategy". The results also showed that 43% of the students partially agreed that "sustainability is always a clear and easy concept for everyone to understand". In contrast, 60.7% of the students did not agree that "sustainability is only a business or marketing strategy because it is fashionable". The statement that "sustainability is fostered by environmental groups that cannot implement it" had relatively even ratings: 38.3% strongly agreed, 34.5% partially agreed and 27.2% slightly agreed.

In a deeper analysis of the responses of students from the different faculties, significant differences were found in the assessments of the statement that "Sustainability is a concern of consumers, which will change the way of producing" ($p = 0.01$). Through the comparison of pairs, a difference was found between the students of Economic Sciences and those of Social Sciences ($p = 0.002$). Those in Economic Sciences strongly agreed with the statement, while those in Social Sciences gave only partial or little agreement.

Subsequently, the respondents were asked to rate their "own knowledge about sustainability". A total of 38.3% of the students rated it "Abundant", 49.0% considered it "Sufficient", and 12.7% "Scant". Upon checking whether the self-assessment of the students' own knowledge was similar or not across the different faculties, we found a significant difference ($p < 0.0001$). In particular, through the comparison of pairs, we found that the students of Economic Sciences and Social Sciences did not differ in their assessment ($p = 0.19$), but the students of Agricultural Sciences did ($p < 0.0001$), as they rated their own knowledge higher than did the other two groups of students.

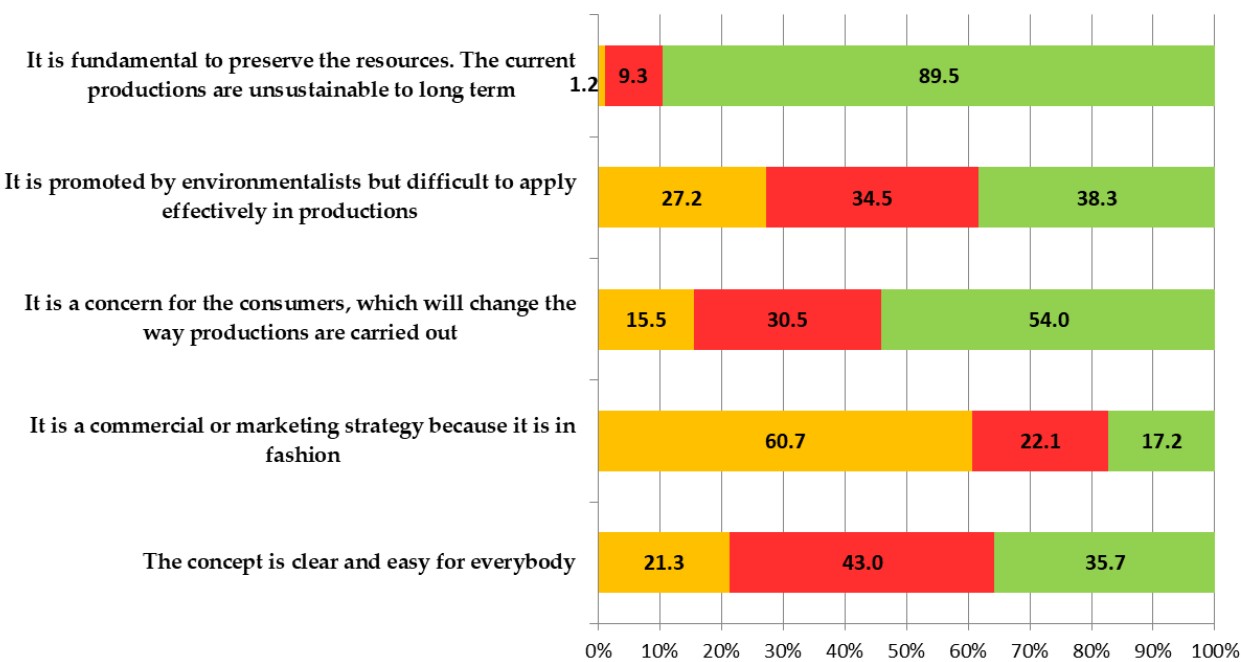

**Figure 6.** Levels of agreement with ideas of Sustainability (%).

The sources of information used by the students for the subject Sustainability (Figure 7) were found to be varied. The main sources of information were as follows: internet search engines (67.8% of students); social networks (56.6%) at a certain distance; specific texts on the subject (33.3%); friends or relatives (27.9%); and traditional media such as television, radio or newspapers (22.6%). Surprisingly, the specific and non-specific courses of their university programs were relegated to the last positions (18.1% and 12.3% of students, respectively), and only 8.4% declared that they were not informed about the subject.

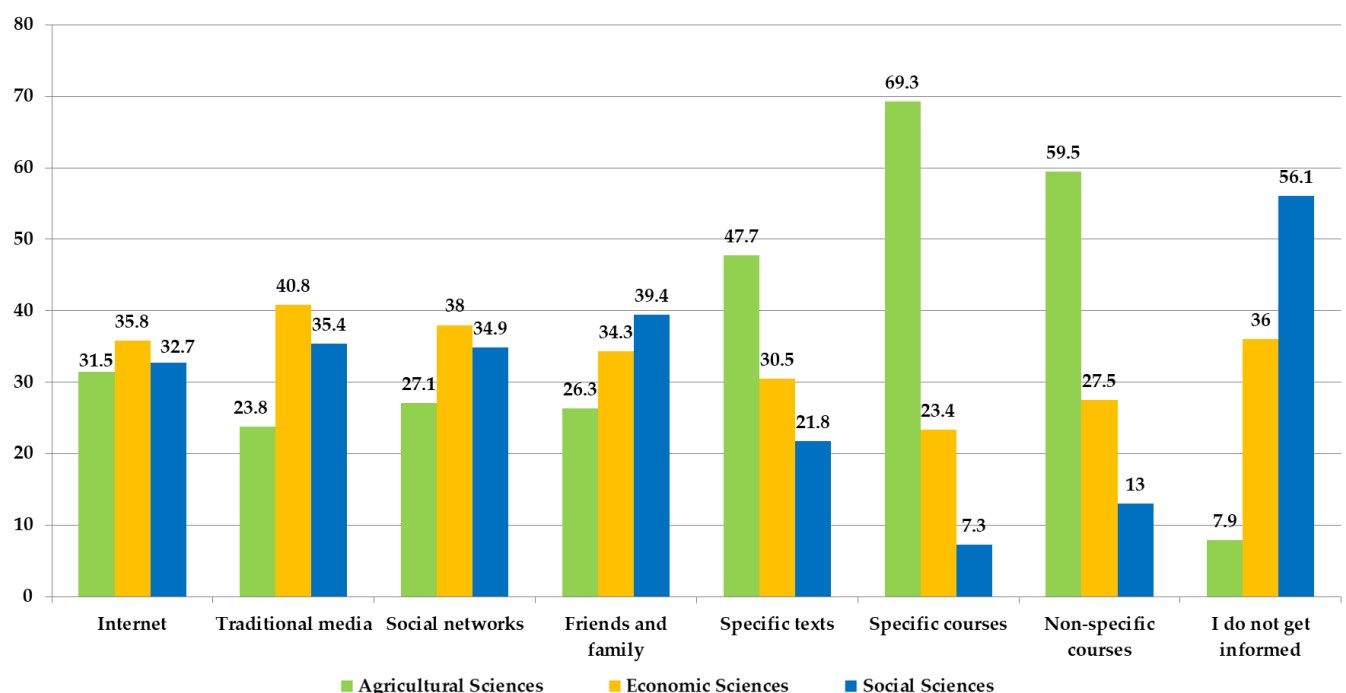

**Figure 7.** Sources from which students obtain information (%).

No significant differences were found between the students of the three faculties, who all reported that their sources of information were internet search engines ($p = 0.05$), and friends and family ($p = 0.12$), but significant differences were found for the other sources of information used ($p < 0.05$). In the comparison of pairs, the students of Agricultural Sciences were those most informed through specific texts, and the specific and non-specific courses of their study program, compared with students of the other two faculties. The students of Economic Sciences were mostly informed through traditional media and social networks. On the other hand, Social Sciences students were the least informed about the subject overall.

Finally, when students were asked to assess the importance of the theme Sustainability for their future work, 77.5% valued it as very important, 14.5% as moderately important, and 8.0% as of little importance for their future work (Figure 8).

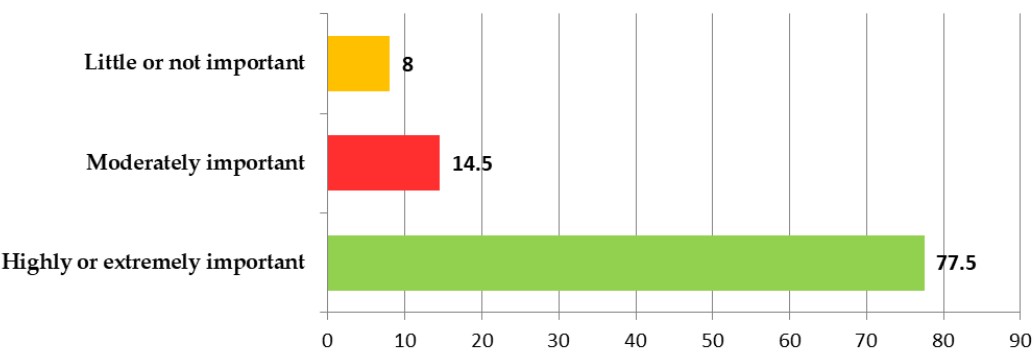

**Figure 8.** Importance of Sustainability for students' future work (%).

The students of the different faculties differed significantly ($p < 0.0001$) in the importance assigned to sustainability for their future work. According to the pair comparison tests, the students of Agricultural Sciences differed from those of Economics ($p = < 0.0001$) and Social Sciences ($p < 0.0001$). Similarly, there were differences between the students of Economic Sciences and Social Sciences ($p < 0.0001$). In summary, students of Agricultural Sciences valued Sustainability as highly or extremely important for their future work, while Social Sciences students and those of Economic Sciences both considered the topic to be of moderate or little importance, or not important at all.

## 4. Discussion

The students and future professionals of the different faculties studied (Agricultural, Social and Economic Sciences) express themselves with a certain uniformity, without substantial differences in terms of the words used in connection with sustainability. Their discourse focuses mainly on terms related to environmental issues (i.e., environment, ecology, care, natural, friendly, planet, good, and pollution), followed by social issues (i.e., responsibility, awareness, life, needs, and well-being) and, lastly, economic issues (i.e., economy and capacity). These findings are consistent with studies reported in the international literature that have been concerned with investigating the perception and knowledge of sustainability of students from different courses of study [5,11,12,25]. Topics such as climate change, environmental protection, conservation of natural resources, sustainable production, and recycling are more present in the minds of the students surveyed compared with the other dimensions of sustainability. Issues related to the environment, such as climate change, droughts, floods, etc., are extensively presented in the media. For this reason, although the students have different academic backgrounds, they are all familiar with these concepts, as also indicated by Burkhart et al. [11]. All respondents in this study are familiar with the term "sustainability", in contrast to results previously reported in the literature [5,12,23]. Through an analysis of the students' discourse, it is possible to highlight a positive trend in the interviewees' familiarity with sustainability, even if it mainly refers to the environmental sphere. This may be due to the growing importance

given in recent years by researchers and the media to the issue of global warming and the need for sustainable development.

On the other hand, knowledge of the social and economic dimensions is marginal. These two dimensions arose spontaneously in only 20.1% and 9.8%, respectively, of the total student discourse. These dimensions were mentioned to a slightly greater extent when specifically requested. Through the 12 proposed statements, substantial differences arose in the importance attributed to each of the dimensions. This suggests that there is no real knowledge of sustainability overall. The results are not surprising as they are in line with previous findings in the literature [5,11,12,15].

In the present work, the hypothesis formulated at the beginning of the investigation was not verified; that is, that students of different academic choices would present a higher degree of knowledge in the area corresponding to their own dimension of sustainability, as proposed by Zwickle et al. [4].

All the interviewed students have high knowledge of the environmental dimension, and sustainability is highly linked to this area. Only 10% of them know that sustainability is made up of three dimensions.

Despite this, a majority of students declared that they had a sufficient or abundant level of knowledge about sustainability. Although this self-assessment appears honest, such overestimation of personal knowledge amongst Argentinian students may be linked to their familiarity with environmental concepts previously reported by the media in their coverage of environmental issues, as has been found in other studies [11], with their lack of knowledge of the existence of the other components caused by a lack of awareness of specific technical concepts. In this regard, it should also be noted that more than half of the interviewees (64.3%), think that the concept of sustainability is neither clear nor easy for everybody to understand.

It can be shown that knowledge of sustainability is not acquired in the academic field, but rather comes from personal training. The same students reported being trained through general information sources and not through specific courses offered by the faculties, except in some specific cases related to the Faculty of Agricultural Sciences.

Notwithstanding the above, the majority of the students surveyed believe that sustainability is not a commercial or marketing strategy and that it could be applied effectively to production. For these reasons and due to its relevance, the topic of sustainability can be seen as important for their professional future, particularly for students of Agricultural Sciences and Economic Sciences. In some studies, carried out in other settings, young people with high academic training recognize the importance of sustainability, consider sustainability to be a key aspect of their university education, and firmly believe that it can bring job opportunities [11,21]. The present work demonstrates that similar levels of interest and awareness are emerging amongst Agricultural Sciences students in Argentina.

## 5. Conclusions

The results of this study increase knowledge about a subject that is still rarely investigated in Argentina. At the academic level, it is important to reflect on the need to improve students' training in sustainability in a more systemic and holistic way. Our results show that, in the minds of contemporary students, the concept of sustainability is highly linked to the environment. Sustainability mainly means preserving natural resources, favoring biodiversity, reducing environmental risks, and finding a balance between the development of humanity and care for the environment. A creative, comprehensive, and multidisciplinary vision will be necessary in the future, which will include knowledge of the three components of sustainability. It is certainly important for all future professionals to have knowledge of environmental sustainability, but it is equally important to increase the knowledge of the other two components, with specificities related to different courses of study.

Environmental sustainability, which entails responsibility in the use of natural resources to guarantee that all generations (present and future) have a good quality of life,

must embrace both economic and social sustainability in a synergic and systemic way. Economic sustainability concerns the ability to generate income and work, and social sustainability guarantees safety, health, justice, and wealth for all populations.

An emphasis on environmental sustainability which leaves aside one of the other two pillars results in an incomplete and abstract concept that is difficult to pursue in the long term. The market must demand sustainable products and bear the totality of production costs, enjoying the positive externalities produced. As with any economic activity, it must generate food safety and good quality of life.

Our findings indicate that knowledge should be improved in the different faculties under study. It is necessary to integrate the topics in a deeper way to provide students with better tools to fulfil their global responsibilities in their future professional work.

Further research should investigate the curricula of the three programs, to inquire whether inadequate student knowledge is due to a lack of specific courses or to their own understanding. This question highlights one of the limitations of the present work, since it did not focus on identifying what type of information students receive, both inside and outside the university. It was limited to inquiring only about the sources from which they obtained their information. Our results indicate that knowledge of the subject is relatively low, which should call into question the quantity and/or quality of the information received. Perhaps it would be interesting to analyze the content of the courses that deal with sustainability, to consider their shortcomings, and identify those topics that should be incorporated into university curricula or specific courses. If such additions are necessary, as suggested by Watson et al. [15], they should be balanced in terms of the three dimensions of the subject for the benefit future professionals who will find themselves making decisions affecting different areas of society.

Argentina is one of the world's leading producers of raw materials. It will be important for professionals involved in sustainability to be highly trained in the subject, to apply its concepts and improve production, within the framework of the three sustainability components.

**Author Contributions:** Conceptualization: J.D.P. and A.B.D.; methodology: J.D.P. and A.B.D.; software: A.B.D.; validation: A.B.D., J.D.P. and J.M.A.; formal analysis: A.B.D., J.D.P. and J.M.A.; investigation: A.B.D., J.D.P. and J.M.A.; resources: J.D.P. and A.B.D.; data curation: A.B.D.; writing—original draft preparation: A.B.D. and J.D.P.; writing—review and editing: J.D.P., A.B.D. and J.M.A.; visualization: A.B.D.; supervision: J.D.P.; project administration: J.D.P. and A.B.D.; funding acquisition: A.B.D., J.D.P. and J.M.A. All authors have read and agreed to the published version of the manuscript.

**Funding:** The present study has been carried out in the framework of the Project "Demetra" (Dipartimenti di Eccellenza 2018–2022, CUP_C46C18000530001), funded by the Italian Ministry for Education, University, and Research.

**Informed Consent Statement:** Informed consent was obtained from all subjects involved in the study.

**Data Availability Statement:** The data presented in this study are available on request from the corresponding author.

**Acknowledgments:** We would like to thank both academic institutions that allowed this collaboration. Moreover, the authors greatly acknowledge the support of anonymous reviewers for their fundamental help in improving the quality of our manuscript.

**Conflicts of Interest:** The authors declare no conflict of interest. The founding sponsors had no role in the design of the study; in the collection, analyses, or interpretation of data; in the writing of the manuscript; or in the decision to publish the results.

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
