# Peer review of "What Does Sustainability Mean? Perceptions of Future Professionals across Disciplines"

_sustainability, doi:10.3390/su14159650_

Round 1
Reviewer 1 Report
The authors of this article explore the very important topic of sustainability. It is extremely important in view of the ever-changing climate.
The introduction is written in a coherent and logical manner. It cites many sources indicating what sustainability is and how it is understood by students in different countries. The aim of the article and the hypothesis are formulated correctly. The sampling method and the presentation and discussion of the tool were done correctly.
The analysis of the collected empirical material is coherent, logical and correctly described. The significance coefficient p needs to be improved to two decimal places, not four. The discussion is presented correctly and clearly. The conclusions are logical and consistent with the research results presented.
The number of sources used is sufficient.
Author Response
Dear Reviewer,
Thank you for your kind comments and suggestions.
Where possible, the p-value has been modified by approximating the value to only 2 decimal places.
In some cases, this was not possible, e.g. < 0.0001, where the p value remained with 4 decimal places.
Best Regards
MEng. Andrea Beatriz Damico
Reviewer 2 Report
The topic of this article is very interesting and will also arouse my interest for knowledge of sustainability of university students. Below are a few observations on the study conducted. Some shortcomings are also mentioned in the observations.
1. It is suggested in the Materials and Methods discussion of CAWI methodology. The authors need to explain their method more clearly and why you choose this method in your research.
2. The discussion and conclusions appear to be brief. The results of the research questions can be further explained and this should be highlighted given the title of the article.
3. The conclusion appears insufficient and should connect more closely to the authors research questions that form the basis of the investigation.
Author Response
Dear Reviewer,
Thank you for your comments and suggestions.
We have tried to improve the work following your suggestions.
In particular:
- It is suggested in the Materials and Methods discussion of CAWI methodology. The authors need to explain their method more clearly and why you choose this method in your research.
We have explained why we chose the CAWI methodology
- The discussion and conclusions appear to be brief. The results of the research questions can be further explained and this should be highlighted given the title of the article.
We have deepened the discussion of some results trying to highlight those most related to the title
- The conclusion appears insufficient and should connect more closely to the authors research questions that form the basis of the investigation.
We have expanded the conclusions trying to make them more relevant to the research question
Best Regards
MEng. Andrea Beatriz Damico